# Effects of Key Components on the Antioxidant Activity of Black Tea

**DOI:** 10.3390/foods12163134

**Published:** 2023-08-21

**Authors:** Weiwei Wang, Ting Le, Wei Wang, Luting Yu, Lijuan Yang, Heyuan Jiang

**Affiliations:** Key Laboratory of Biology, Genetics and Breeding of Special Economic Animals and Plants, Ministry of Agriculture and Rural Affairs, Tea Research Institute, Chinese Academy of Agricultural Sciences, Hangzhou 310008, China; wangwei11211@tricaas.com (W.W.); letin@tricaas.com (T.L.); ww1040491839@163.com (W.W.); yuluting@tricaas.com (L.Y.); yanglijuan@tricaas.com (L.Y.)

**Keywords:** black tea, antioxidant activity, catechins, oxidation polymer, theabrownin

## Abstract

Many components (such as tea polyphenols, catechins, theaflavins, theasinensins, thearubigins, flavonoids, gallic acid, etc.) in black tea have antioxidant activities. However, it is not clear which components have a greater influence on the antioxidant activity of black tea. In this study, the antioxidant activity and contents of tea polyphenols, catechins, theaflavins, thearubigins, theabrownins, TSA, total flavonoids, amino acids, caffeine, and total soluble sugar were analyzed in 51 black teas. Principal component analysis (PCA), orthogonal partial least-squares discrimination analysis (OPLS-DA), and the correlation analysis method were used for data analysis. The results showed that catechins in tea polyphenols were the most important components that determine the antioxidant activity of black tea. Among them, epicatechin gallate (ECG), epi-gallocatechin gallate (EGCG), epicatechin (EC), and epi-gallocatechin (EGC) were significantly positively correlated with the antioxidant activity of black tea, and theabrownin was negatively correlated with the antioxidant activity of black tea. Furthermore, this study analyzed the correlation between the changes in catechin and its oxidized polymers with antioxidant activity during black tea fermentation; it verified that catechins were significantly positively correlated with the antioxidant activity of black tea, and theabrownin showed a negative correlation. And the antioxidant activity of catechins and their oxidation products in vitro and their correlation in black tea processing were used as validation. This study provides a comparison method for comparing the antioxidant activity of black tea.

## 1. Introduction

In the year 2020, the global tea production was 6.27 million tons, and it had been growing for 10 years (data source: China Tea Marketing Association). Black tea is the most produced and consumed tea in the world and accounts for over 70% of the world’s tea production. China, India, Kenya, and Sri Lanka are the world’s largest producers of black tea [1]. The global tea export and price in major black-tea-producing countries such as Kenya and Sri Lanka have shown fluctuations. In the past five years, black tea has accounted for 10–15% in the six major Chinese tea categories, and the proportion is gradually increasing; all 18 tea-producing provinces in China also produce black tea. Congou black tea is an important type of Chinese black tea, which shows variety and a wide distribution. The manufacturing process of Congou black tea includes withering, rolling, fermentation, and firing, as fermentation is the key process [2].

Oxidation is a process in which the body uses oxygen; the body continuously produces reactive oxygen species (ROS). More ROS will disrupt intracellular metabolic balance. It is crucial for health to find safe and effective natural antioxidants [3]. Black tea has good antioxidant, anticancer, anti-microbial, and anti-obesity activities [4]. Antioxidant activity is researched mostly, and the methods to evaluate the antioxidant activity of tea mainly include ferric reducing antioxidant power (FRAP), 1,1-dipheny1-2-picrylhydrazyl (DPPH), 2,2′-azinobis-(3-ethylbenzthiazoline-6-sulphonate (ABTS), and Oxygen Radical Absorbance Capacity (ORAC). In many previous studies, the ORAC of 52 black tea samples from different geographical locations and of different plantation elevation and leaf grade were found to be in the range of 548.8 μmol TE/g to 2824.4 μmol TE/g [5]. The FRAP, total reducing capacity assays, and DPPH methods that were used to analyze the same tea sample obtained different results; the leaf grade and different processing technologies also influenced the antioxidant activity of black tea [6,7,8].

Some studies showed that polyphenols, especially catechins, theaflavins, thearubigins, gallic acid, and flavonoids, were the main antioxidant active components of black tea [9,10,11,12,13]. Most polyphenols show a decreasing trend during black tea processing; catechins are oxidized to form theaflavins, thearubigins, and theabrownins [14]. The series of oxidation reactions during the fermentation process leads to significant differences in the antioxidant active ingredients in black tea. However, there are few comparative studies on the antioxidant activity of different black teas. Factors such as processing technology, altitude, shooting period, tea grade, and origin can affect the antioxidant activity of black tea [15,16,17]. Many studies have illustrated the components of black tea with antioxidant activity, but the analysis of components related to antioxidant activity among different black teas is not clear. This study aimed to investigate the differences in the antioxidant activity and chemical composition of black tea from different sources. It analyzed the correlation between the main components and antioxidant activity and discussed the effects of catechins and their oxide polymers on antioxidant activity, which were verified using in vitro tests of pure compounds and samples from black tea processing.

## 2. Material and Methods

### 2.1. Experimental Materials

A total of 51 black teas were analyzed in this study. Among them, 47 were bought from Yunnan, Sichuan, Fujian, Guizhou, Guangdong, Anhui, Jiangxi, Hubei, Shanxi, Hunan, and Henan Provinces, 3 black tea samples were from Sri Lanka, and 1 black tea sample was from India. Longjing 43 tea leaves were selected for the purpose of this study, following the picking standards of collecting a bud with two leaves. The tea leaves were collected in July 2021 in Shengzhou (120°49′ E; 29°35′ N) (Zhejiang, China).

Eight catechin standards (≥98%), including (+)-catechin (C), (−)-epicatechin (EC), (−)-gallocatechin (GC), (−)-epi-gallocatechin (EGC), (−)-catechin gallate (CG), (−)-epicatechin gallate (ECG), (−)-gallocatechin gallate (GCG), and (−)-epi-gallocatechin gallate (EGCG), and caffeine (CAF) and gallic acid (GA) were purchased from Sigma Aldrich Shanghai Trading Co., Ltd. (Shanghai, China). Four theaflavins standards (≥98%), including Theaflavin (TF), Theaflavin-3-gallate (TF-3-G), The-3′-gallate (TF-3′-G), and Theaflavin-3-3′-gallate (TF-3,3′-G), were purchased from Wako Pure Chemical Corporation (Tokyo, Japan). Theasinensin A (TSA) (≥98%) was provided by Nagasaki University, Japan.

Thearubigin and theabrownin were separated and prepared by the laboratory: Crush–tear–curl (CTC) broken black tea was extracted by boiling water, then filtration and cooling, the same volume of ethyl acetate was used to extract, shaking vigorously for 5 min, and standing, the water layer was extracted twice with the same volume of n-butanol, the n-butanol phase was rotated and evaporated to dryness to obtain the crude product of TRs, the water layer was further extracted with dichloromethane to remove caffeine and chlorophyll, and the remaining water layer was rotated and evaporated to obtain TBs product.

Chromatography-grade solvents, including acetonitrile (ACN), ethyl acetate (EA), phosphoric acid (PA), and glacial acetic acid (GAA) were purchased from Merck (Darmstadt, Germany). Analytical reagents, including sodium carbonate, glutamate, quercetin rhamnoside, ninhydrin, stannous chloride, oxalic acid, disodium hydrogen phosphate, potassium phosphate monobasic, anthrone, sulfuric acid, n-butanol, ethanol, methanol, foline-phenol, and ethyl acetate were purchased from Shanghai Macklin Biochemical Co., Ltd. (Shanghai, China). The total antioxidative capacity (T-AOC) assay kit, which was used to determine the tea sample, was purchased from Nanjing Jiancheng bioengineering institute, and the total antioxidative capacity (T-AOC) assay kit, which was used to determine the monomers, was purchased from Beyoyime.

### 2.2. Black Tea Processing and Samplings

(1) Withering: the fresh tea leaves (FTL, 26 kg) were spread on circular net plates (leaf thickness of each plate was 2–3 cm) for 21 h at room temperature, until the moisture level reached 62–64% (Figure 1).

(2) Rolling: the withering tea leaves (WTL) were subjected to rolling for 60 min in a rolling machine (Figure 1).

(3) Fermentation: the rolling tea leaves (RTL) were put into intelligent, artificial climate box with fermentation environment temperatures of 23 °C, and the humidity was 95% (Figure 1).

(4) Samplings: 100 g of each treatment was taken individually for 0 h, 1 h, 2 h, 3 h, 4 h, 5 h, 6 h, and 7 h during fermentation, then all of them were stored in liquid nitrogen. These samples were vacuum freeze-dried in an LGJ-50C freeze dryer.

### 2.3. Samples Preparation and Extraction

The tea samples were broken up by a grinder twice, and the grinding parameters were 20,000 rpm and 30 s. Then, 0.125 g tea powder was weighed (accurate to 0.0001 g) into 10 mL centrifuge tubes, and 5 mL 70% aqueous methanol was added, extracted for 10 min at 70 °C; after centrifugation at 3500 rpm for 10 min, the supernatant was collected into 10 mL volumetric flask, and the solid was re-extracted by the above process, then two supernatants were combined and diluted with 70% methanol into 10 mL, then diluted by 50 times with 70% methanol as the sample extract.

### 2.4. Determination of Total Antioxidant Activity (FRAP Method)

Referring to the instructions of the total antioxidative capacity (T-AOC) assay kit, reagent 1, sample extracts (0.2 mL), reagent 2, reagent 3, and reagent 4 were added as required. The contrast was set as the addition time of sample extracts (0.2 mL) to the last one, just after the reagent 4 addition for the termination of reaction. Definition: every 0.01 increase in the absorbance (OD) value of the reaction system per milliliter of black tea extract per minute was a total antioxidative unit at 37 °C.

### 2.5. Determination of Total Polyphenols (TP)

A total of 1 mL sample extract was added to 20 mL test tube, 5 mL 1:10 diluted Folin–Ciocalteu reagent was added, then 4 mL sodium carbonate solution (75 g/100 mL) was added after 3–8 min. After incubation at room temperature for 60 min, the absorbance of the mixture was measured at 765 nm using spectrophotometer (UV 3600, Shimadzu, Japan). Gallic acid was used as the standard for a calibration curve; results were expressed as mg/g dry sample.

### 2.6. Determination of Catechins, TSA, GA and CAF

The analysis was carried out by a HPLC (Shimadzu, Japan), with Nacalai Tesque C18-AR-II column (5 μm, 250 × 4.60 mm, Cosmosil, Kyoto, Japan); the mobile phase A contained 50 mmol/L phosphate in aqueous solution and mobile phase B was pure acetonitrile; the column temperature was maintained at 35 °C; the detection wavelength was 280 nm; the flow rate and injection volume were 800 uL/min and 10 μL, respectively; and the gradient was set as follows: 0 min, 4%B, 0–39 min linear gradient increased to 30% B, 39–54 min linear gradient increased to 75% B, and 54–55 min decreased to 4% B. Results were expressed as mg/g dry sample.

### 2.7. Determination of Theaflavins

The analysis was carried out by a HPLC (Waters, Milford, MA, USA), with Elite C18 column (5 μm, 250 × 4.60 mm, Elite, Dalian, China); the mobile phase A contained 2% glacial acetic acid phosphate in aqueous solution and mobile phase B contained acetonitrile and ethyl acetate mixed 7:1; the column temperature was maintained at 35 °C; the detection wavelength was 380 nm; the flow rate and injection volume were 1.5 mL/min and 10 μL, respectively; and the gradient was set as follows: 0 min, 8% B, 0–27 min linear gradient increased to 26% B, 27–30 min linear gradient decreased to 8% B, and 30–35 min maintained 8% B. Results were expressed as mg/g dry sample.

### 2.8. Determination of Free Amino Acids

A total of 1 mL sample extract together with 0.5 mL buffer of pH 8 and 0.5 mL ninhydrin solution were added to 25 mL volumetric flask successively, then the reaction liquid heated at 100 °C for 15 min using water bath, diluted with aqueous solution to the volume after cooling, and the absorbance of the mixture was measured at 570 nm using spectrophotometer. Glutamate was used as the standard for a calibration curve; results were expressed as mg/g dry sample.

### 2.9. Determination of Total Sugar [18]

A total of 1 mL sample extract and 4 mL sulfuric acid with anthrone (2 g/L) were added to test tube with stopper, then the reaction liquid heated at 100 °C for 10 min using water bath, placed in ice bath for 10 min immediately, and then the absorbance of the mixture was measured at 620 nm using spectrophotometer. Glucose was used as the standard for a calibration curve; results were expressed as mg/g dry sample.

### 2.10. Determination of Flavonoids

A total of 0.5 mL sample extract and 10 mL aluminum trichloride in aqueous solution (1.76 g/100 mL) were added to the test tube; after incubation at room temperature for 10 min, the absorbance of the mixture was measured at 420 nm using spectrophotometer. Quercetin rhamnoside was used as the standard for a calibration curve; results were expressed as mg/g dry sample.

### 2.11. Determination of Thearubigins (TRs) and Theabrownins (TBs) [18]

TR and TB contents were quantified through systematic analysis. A total of 3 g of tea sample was infused with boiled pure water and heated at 100 °C for 10 min using water bath, then the aqueous extract was filtered. A total of 30 mL tea-cooled extracts was put into a 60 mL separating funnel, and the same amount of ethyl acetate was added, the mixture was shaken for 5 min, then placed still, the water layer was discharged, and the ethyl acetate layer was poured out, 2 mL ethyl acetate layer was absorbed, and 95% ethyl alcohol was added to obtain solution A of 25 mL. A total of 2 mL water layer was absorbed, and 2 mL saturated oxalic acid solution and 6 mL of pure water were added, then 95% ethyl alcohol was added to obtain 25 mL solution B. A total of 15 mL ethyl acetate layer was put into a 30 mL separating funnel, and the same amount of sodium bicarbonate solution (2.5 g/100 mL) was added, the mixture was shaken for 30 s, then placed still, 4 mL ethyl acetate layer was absorbed, and 95% ethyl alcohol was added to obtain 25 mL solution C. A total of 15 mL tea-cooled extract was put into a 30 mL separating funnel, 15 mL of n-butyl alcohol was added, the mixture was shaken for 3 min, then placed still, the upper layer was discarded, 2 mL water layer was absorbed, and 2 mL saturated oxalic acid solution and 6 mL pure water were added, then 95% ethyl alcohol was added to obtain 25 mL solution D. The absorbance of four solutions were measured at 380 nm and recorded as EA, EB, EC, and ED. The contents were calculated using the following formulas: TRs (%) = 7.06 × (2 EA + 2 EB − EC − 2ED); TBs (%) = 2 × ED × 7.06 (Wang et al., 2021); results were expressed as mg/g wet sample.

### 2.12. Statistical Analysis

All the data were recorded as mean ± standard deviation (SD) of three replicates. The analysis of significant differences between the means was carried out by one-way analysis of the variance (ANOVA), followed by the Duncan test to compare the means for significant variation (*p* < 0.05). The supervised orthogonal partial least squares discriminant analysis (OPLS-DA) was used to investigate the tea samples using SMICA 14.0 software for multivariate analysis.

## 3. Results and Discussion

### 3.1. Variable Coefficient Analysis of Antioxidant Activity and Chemical Compounds of Black Tea

In this study, all spectrophotometric measurements were made on the same instrument (UV-3600 Shimadzu, Kyoto, Japan), and catechins and their dimers were detected by high-performance liquid chromatography (LC-20 Shimadzu, Japan). T-AOC (FRAP principle) and chemical components of 51 black tea samples were analyzed. T-AOC was found to be in the range of 128.36 U to 930.16 U, and the variable coefficient was 0.36. The coefficient of variation reflects the statistical dispersion of a set of data; the range of coefficient of variation is 0–1, and the larger coefficient shows a greater difference of the data. As shown in Table 1, the four catechins (EC, EGC, ECG, and EGCG) had the largest variable coefficient (0.59–0.95), followed by the theaflavins (0.31–0.66), and the amino acids and caffeine had the smallest coefficient of variation, which indicated that the catechins of black tea from different sources had the biggest differences. The maximum were several times higher than the minimum contents, ranging from 1.92 times (AA) to 159.22 times (ECG), and the maximum contents of four phenotypic catechins and TSA contents were 10 times more than the minimum contents. This result was also consistent with the previous study [19]. On the whole, the oxidation reaction of catechins during black tea fermentation led to a significant decrease in the content of catechins; it was the most variable component among black teas due to different varieties and processing technology [5].

### 3.2. Differential Compounds Analysis of Black Tea with Different Antioxidant Activity

Principal component analysis is used to reflect the abundance of metabolites; the closer position in the score plot, the more similar the compounds and contents are, and the load diagram plot reflects the distribution of metabolites in different principal components of the samples. There were significant differences in antioxidant activity of black tea from different sources [20]. In this study, the supervised OPLS-DA analysis found that tea samples with high T-AOC (480–930 U) were distributed in quadrants 1 and 4 and located on the right of the *x*-axis, tea samples with low T-AOC (128–300 U) were distributed in quadrants 2 and 3 and located on the left of the *x*-axis, and tea samples with intermediate T-AOC (300–480 U) were distributed in the middle and above the second quadrant (Figure 2a). As shown in Figure 2, the first eight important variables for the classification of tea were TP, ECG, TSA, EGCG, GA, EC, CAF, and AA (VIP > 1 and *p* < 0.05) (Figure 2c); these compounds were used to distinguish black tea samples with different antioxidant activities. There were eight compounds distributed at the far right end of the Load diagram (Figure 2b). The analysis results were available; two principal components explained 44.89% of the total variations, and the first principal components explained 38.65%. The cross-validation results showed no over-fitting of the model with 200 iterations (intercepts, R2 = 0.047 and Q2 = −0.213) (Figure 2d).

To show the antioxidant activities of 51 black tea samples more clearly, a heat map of eight differential compounds was constructed (Figure 2e), in which the red block represented high contents and the blue block represented low contents. Interestingly, the contents of all eight differential compounds exhibited higher contents in high antioxidant-activity tea samples. The tea samples investigated were clearly divided into two groups: the high antioxidant-activity and some moderately antioxidant-activity tea samples in class I, and the low antioxidant activity and the remaining moderately antioxidant-activity tea samples in class II.

In many recent studies, catechins, theaflavins [21], TRs [22], TBs [23], GA [24], and flavonoids [25] were the antioxidant activity compounds in black tea. Theaflavins had higher potential of antioxidant activity than EGCG [26]. In this study, theaflavins, TRs, TBs, and flavonoids were not the differential compounds; TP and catechins were the differential compounds. Although AA and CAF had weak antioxidant activity, they were differential compounds, and the reason might be that the contents of AA and CAF in tea samples were lower due to a higher maturity, which was similar to that of catechins.

### 3.3. Correlation Analysis of Antioxidant Activity and Chemical Compounds of Different Black Teas

Based on the results of the correlation analysis, Figure 2a showed that the compounds of ECG, TPs, EGCG, TSA, GA, EC, TF-3,3′-G, EGC, CAF, and AA had a significant positive correlation with antioxidant activity, and they are listed in the descending order of correlation coefficient. Interestingly, the 10 positive-correlation compounds contained all eight differential compounds, and positive correlation components and differential compounds reached a high consistency, whilst the VIP value of the remaining two components was also high. Phenolic compounds, especially catechins, were the potential antioxidant compounds of black tea [27,28], and catechol and pyrogallol-type catechins had stronger antioxidant activity [29]. The phenotypic catechins had a stronger free-radical scavenging ability than non-phenotypic catechins [30,31]. This research showed that four phenotypic catechins were positively correlated with the antioxidant activity of black tea, the correlation coefficient of ECG and EGCG with black tea antioxidant activity were also higher than EC and EGC, and the phenomenon was similar to the previous studies [32]. CAF and AA were also had a significant positive correlation with antioxidant activity, which could be due to them being significantly positively correlated with those compounds with higher antioxidant activities, such as TPs, GA, EGCG, ECG, and TF-3,3′-G.

The oxide products of catechins, such as theasinensins, theaflavins, thearubigins, and theabrownins were also important phenolic compounds with antioxidant activity in black tea [33], however only TSA and TF-3,3′-G were positively correlated with the antioxidant activity of black tea. The previous studies also showed that TF-3,3′-G was the strongest antioxidant component among theaflavin compounds [5]. This might be related to the stability of theaflavins, which were less stable than catechins, but TF-3,3′-G was more stable in four theaflavin components [34]. This study found that TSA had a higher correlation coefficient than theaflavins for the first time. The contents of TF, TF3G, TF3′G, and thearubigins were not significantly correlated with antioxidant activity. On the contrary, theabrownins were significantly negatively correlated with antioxidant activity. In addition, flavonoids and total sugar were not significantly correlated with antioxidant activity.

Phenolics were primary antioxidant compounds of black tea. Figure 3b showed that catechins and their dimers showed a significant positive correlation with the antioxidant activity of black tea, and the correlation coefficient was above 0.9. But the correlation coefficient gradually decreased when thearubigins and theabrownins were increasing, and there was a significant decrease when coupled with theabrownin production. It could be seen that catechins and their oligomers were the primary components that determined the antioxidant activity of black tea. Although TRs and TBs were also important antioxidant components, and their contents were higher than catechins in black tea [13,14], they were not important factors for determining the antioxidant activity of black tea. In particular, the content of tea TBs was significantly negatively correlated with the antioxidant activity. 

### 3.4. Correlation Analysis of Antioxidant Activity and Chemical Compounds during Black Tea Fermentation

The black tea fermentation process is accompanied by a series of oxidation reactions of catechins. Catechins firstly forms theaflavin and theasinensin through the benzotropolone ring and disproportionation. These catechin dimers are unstable and are prone to forming thearubigin and theabrownin through oxidation reactions [35]. As mentioned above, catechins and its oxidized products were important substances that determined the antioxidant activity of black tea. Figure 4 showed that both catechins and antioxidant activity decreased during the fermentation process, and showed a very consistent dynamic change. First, the TSA content increased, then it decreased slowly. Theaflavins showed a rapid increase first, then reached a stable trend. The dynamic changes in TSA and theaflavins were slightly different, and the TSA content reached maximum after 2 h of fermentation, whilst theaflavins reached maximum at 4–5 h. Thearubigins showed a trend of increasing first, then decreasing. Theabrownin increased gradually during the fermentation process. The dynamic changes in catechins, theaflavins, thearubigins, and thearubigins were consistent with previous studies This study firstly studied the change of TSA, and then compared this with theaflavins [2,18].

Based on the results of correlation analysis, Figure 5 showed that catechins and their dimers had a significant positive correlation with antioxidant activity, and the correlation coefficient was above 0.9. The correlation coefficient decreased slightly after the increase in thearubins, and the correlation coefficient decreased significantly after the increase in theabrownins. The analysis of the correlation between antioxidant activity with catechins and their oxidized polymers during the fermentation process of black tea found that the higher degree of catechins oxidation, the lower the correlation with antioxidant activity, and theabrownins were significantly negatively correlated with the antioxidant activity during the black tea fermentation process.

### 3.5. Comparison of Antioxidant Activities of Catechins and Their Oxidation Products

Based on the results of the correlation analysis, the higher the oxidation polymerization degree of catechins, the lower the correlation between the products and antioxidant activity of black tea. The comparison of antioxidant activities among these compounds was shown in Figure 6; the concentration of extract was 1 mg/mL, and four catechins had the highest antioxidant activity. Among them, ECG was the strongest. Even though EGCG was the most studied antioxidant compound in tea, the study showed that ECG had a higher ability to chelate copper ions in the FRAP method [36]. As for the FeSO_4_ equivalent ability, catechin dimers took second place, and TSA was stronger than theaflavins, whilst theabrownins were in last place. It showed similar results with the correlation analysis; ECG was also more prominent than other catechins in differences and correlations, and TSA was more prominent than theaflavins and theabrownins.

## 4. Conclusions

Through the partial least squares discriminant analysis (OPLS-DA), the study found that black tea samples from different sources had significant differences in antioxidant activity, and TP, ECG, TSA, EGCG, GA, EC, CAF, and AA were the differential compounds. There were eight compounds with a significant positive correlation with the antioxidant activity. Among them, TP, ECG, EGCG, and EC decreased during the black tea fermentation process. And the study found that the antioxidant activity of black tea also showed a decreasing trend during fermentation. It was speculated that the antioxidant activity of black tea was weakened with the oxidative polymerization of catechins gradually, and the antioxidant activity of catechins and their oxides in vitro supported this view. TSA was the differential component in catechin dimers, and TSA had a higher correlation coefficient with antioxidant activity than theaflavins and theabrownins, and TSA was also stronger than theaflavins in the function of inhibiting melanin synthesis [37]. TSA increased first, then decreased slowly during the black tea fermentation process, which was similar to theaflavins. CAF and AA had the smallest coefficients of variation, but they were differential components and were significantly positively correlated with the antioxidant activity of black tea. However, they remained relatively stable during the black tea process [18].

The antioxidant activity of black tea was mainly determined by the content of catechins. The correlation coefficient between the sum of catechins and its dimers with the antioxidant activity could reach more than 0.9, and ECG had the highest correlation coefficient, however the correlation coefficient gradually decreased when thearubigins and theabrownins were increasing. The conclusion could also be proved by exploring the changes of antioxidant activity during the fermentation of black tea and the antioxidant activity of catechins and their oxides in vitro; both the antioxidant activity and catechins contents decreased gradually during black tea fermentation, whereas theabrownin content increased gradually. This study provides a good way for evaluating the antioxidant activity of black tea and could provide new ideas for improving the antioxidant activity of black tea samples and developing new products.

## Figures and Tables

**Figure 1 foods-12-03134-f001:**
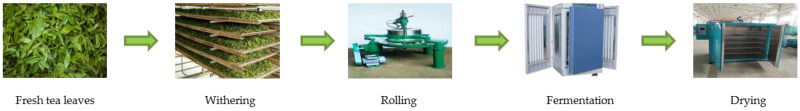
Black tea processing flow chart.

**Figure 2 foods-12-03134-f002:**
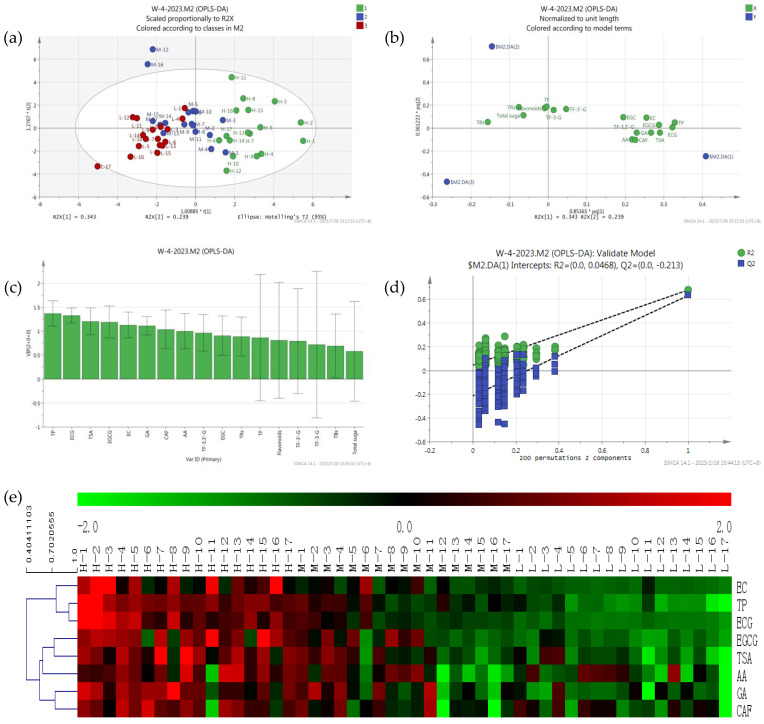
Metabolomics analysis of differential compounds in different black teas: PCA score plot (**a**), Load diagram plot (**b**), VIP plot (**c**), of 51 black teas; Cross−validation plot (**d**) of the PLS−DA model with 200 permutation tests, and heat map (**e**) of differential compounds in 51 black teas.

**Figure 3 foods-12-03134-f003:**
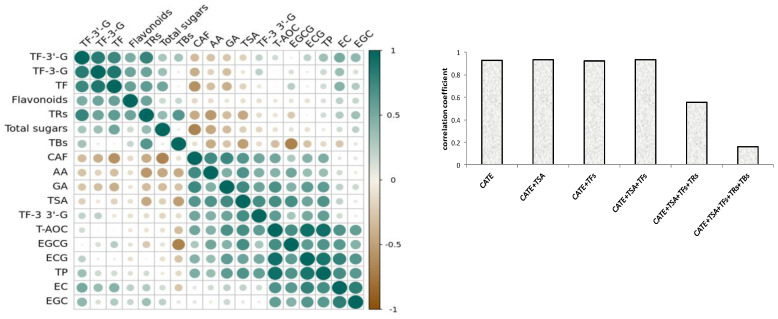
Correlation coefficient plot (**a**) between different components and antioxidant activities of 52 black teas; (**b**) correlation of the sum of catechins and their aggregated products with antioxidant activity of 52 black teas.

**Figure 4 foods-12-03134-f004:**
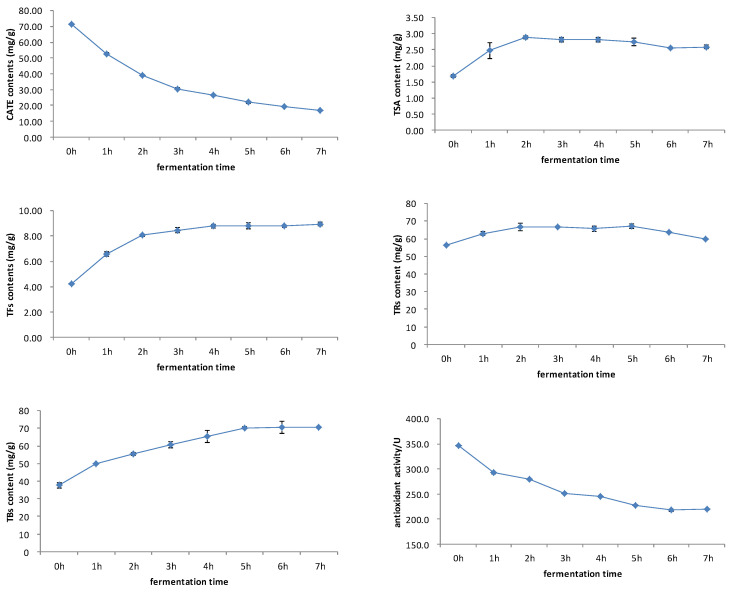
Changes in antioxidant activity, catechins, theaflavins, TSA, thearubigins, and theabrownins during black tea fermentation.

**Figure 5 foods-12-03134-f005:**
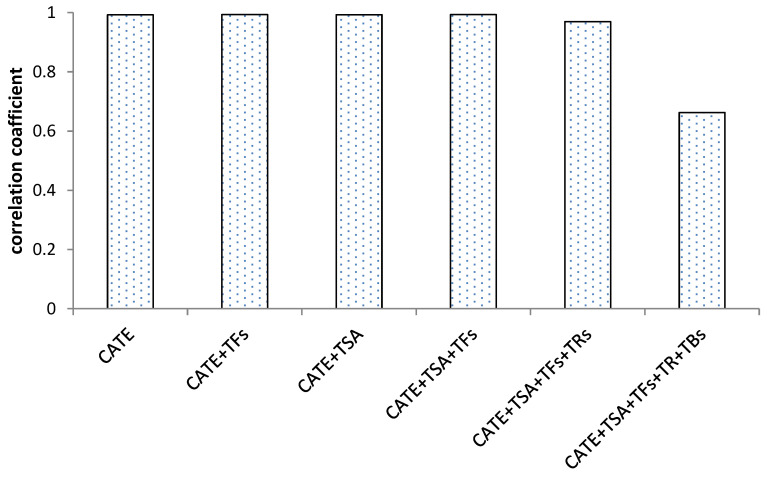
Correlation of the sum of catechins and their aggregated products with antioxidant activity during black tea fermentation.

**Figure 6 foods-12-03134-f006:**
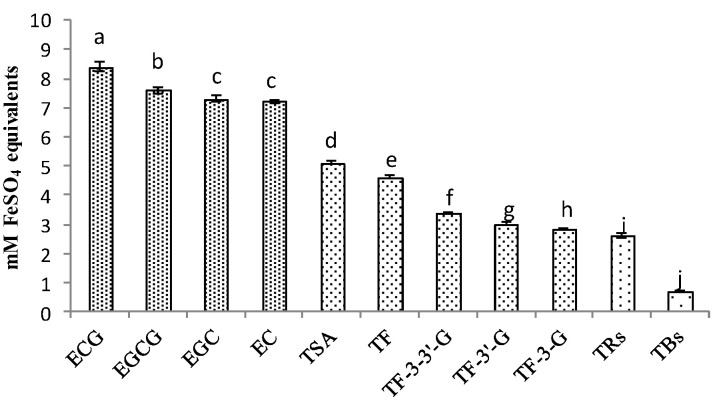
Comparison of antioxidant activities of catechins and their oxidation products. ANOVA was applied with *p*-value; ^a,b,c,d,e,f,g,h,i,j^ different letters above the column indicate significant differences (*p* < 0.05).

**Table 1 foods-12-03134-t001:** Chemical composition distributions of black tea from different regions (mg/g).

Compounds	Minimum Content	Maximum Content	Average Content	Variable Coefficient
EC	0.07	5.19	1.22 ± 1.17	0.96
EGC	0.21	5.85	1.43 ± 1.17	0.82
ECG	0.16	25.66	6.47 ± 5.90	0.80
EGCG	1.40	15.89	7.35 ± 3.82	0.59
TF	0.21	2.65	0.74 ± 0.49	0.66
TF-3-G	0.62	2.72	1.51 ± 0.53	0.35
TF-3′-G	0.40	2.57	1.03 ± 0.38	0.37
TF-3,3′-G	0.98	5.27	3.08 ± 0.96	0.31
TSA	0.27	4.70	2.27 ± 1.02	0.45
TP	52.70	162.31	109.23 ± 21.39	0.20
AA	24.23	46.52	36.63 ± 5.66	0.15
CAF	14.81	36.22	28.89 ± 4.30	0.15
GA	0.54	3.44	2.06 ± 0.64	0.31
Flavonoids	2.73	20.09	13.03 ± 3.20	0.25
Total sugars	124.17	347.73	179.63 ± 45.80	0.26
TRs	25.92	92.29	46.08 ± 14.67	0.32
TBs	43.23	122.14	68.65 ± 19.17	0.28

EC, epicatechin; EGC, epi-gallocatechin; ECG, epicatechin gallate; EGCG, epi-gallocatechin gallate; TF, Theaflavin; TF-3-G, Theaflavin-3-gallate; TF-3′-G, The-3′-gallate; TF-3,3′-G, Theaflavin-3-3′-gallate; TSA, Theasinensin A; TP, tea polyphenols; AA, amino acid; CAF, caffeine; GA, gallic acid; TRs, thearubigins; and TBs, theabrownins.

## Data Availability

The datasets generated for this study are available on request to the corresponding author.

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
