# Peer review of "Effects of Key Components on the Antioxidant Activity of Black Tea"

_foods, 2023, doi:10.3390/foods12163134_

Round 1

Reviewer 1 Report

I am very grateful you for the invitation to review manuscript foods-2520957 by Wang and coauthors "Analysis of key components on the antioxidant activity of black tea”. In this study, the antioxidant activity and content of tea polyphenols, catechins, theaflavins, thearubigins, theabrownins, TSA, total flavonoids, amino acids, caffeine, and total soluble sugar were analyzed in 51 black teas. The work is interesting but needs adjustments to increase the quality of the material.

Comments:

- Abstract: Please indicate a better step-by-step about the work.

- Abstract, “Among them, ECG, EGCG, EC”: Better specify abbreviation on first appearance.

- Specify which conclusions were found according to the sentence “Furthermore, the association of the catechins and their oxidized polymers with black tea was analyzed, and the antioxidant activity of catechins and their oxidation products in vitro and their correlation in black tea processing were used as validation”:

- Abstract: The conclusion is not related to the work objective. Please rewrite the sentence, responding to the objective.

- Keywords: Change the repeated keywords by different words from the title.

- Introduction: Please provide more information regarding the black tea market, including global production and market size ($).

- Introduction: The sentence “FRAP and DPPH [4] and leaf grade [5] also had difference [6]” is generic. Each determination method is specific for different groups of compounds and are complementary. Review the sentence.

- “47 black tea samples from Yunnan, Sichuan, Fujian, Guizhou, Guangdong, Anhui, Jiangxi, Hubei, Shanxi, Hunan, Henan Province, and 3 black tea samples from Sri Lanka, 1 black tea sample from India.”: Review the sentence, information is missing (were used? Analyzed?).

- 2.4. Determination of total antioxidant activity (FRAP method): The use of a single methodology for a study that wants to elucidate antioxidant properties is critical, since each method can detect different chemical groups and components.

- 2.9. Determination of total sugar: Change “spectrophotometer, Glucose” to “spectrophotometer. Glucose”.

- 2.11. Determination of thearubigins (TRs) and Theabrownins (TBs): Change “of 25 mL. the 15 mL ethyl acetate layer” to “of 25 mL. 15 mL ethyl acetate layer”.

- 3.1. Variable coefficient: Change “of a set of data, As shown in Table 1” to “of a set of data. As shown in Table 1”.

- “the oxidation reaction of catechins during black tea fermentation: Deepen the discussion on oxidation.

- In addition to variation by fermentation, what other discussions can be observed in relation to the wide variety of composition of chemical components? The discussion must be deepened.

- Figure 1: The figure resolution is low, and the legend is not visible.

- 3.3. Correlation analysis of: Change “analysis, figure 2.a” to “analysis, Figure 2.a”.

- Figure 2: The figure resolution is low, and the legend is not visible.

- The biochemical aspects of production and degradation of antioxidant compounds must be deepened, given the large variation between samples.

- 3.4. Correlation analysis of antioxidant activity and chemical compounds during black tea fermentation: Similarly, discussion of changes during fermentation should be highlighted.

- Standardize the use of units throughout the text (ml, mL, for example).

Author Response

The manuscript has been modified according to modification suggestions.

Response to Reviewer 1 Comments

Point 1: Abstract: Please indicate a better step-by-step about the work

Response 1: The Abstract were revised according to the suggestion, and specially supplied the analysis method and related correlation analysis.

Point 2: Abstract, “Among them, ECG, EGCG, EC”: Better specify abbreviation on first appearance

Response 2: We revised them, and added the full name of the specify abbreviation on first appearance.

Point 3: Specify which conclusions were found according to the sentence “Furthermore, the association of the catechins and their oxidized polymers with black tea was analyzed, and the antioxidant activity of catechins and their oxidation products in vitro and their correlation in black tea processing were used as validation”:

Response 3: It was modified. The new sentence is “Furthermore, this study analyzed the correlation between changes of Catechin and its oxidized polymer with the antioxidant activity during black tea fermentation. It were found out that Catechins were significantly positively correlated with the antioxidant activity of black tea, Theabrownin showed negative correlation. And the antioxidant activity of catechins and their oxidation products in vitro and their correlation in black tea processing were used as validation. This study provides new idea for comparing the antioxidant activity of black tea.”

Point 4 : Abstract: The conclusion is not related to the work objective. Please rewrite the sentence, responding to the objective.

Response 4: Thanks for the suggestion. We revised the conclusion, and wish the new chapter better.

Point 5 : Keywords: Change the repeated keywords by different words from the title.

Response 5: The keywords:” black tea; antioxidant activity; components; catechins” were changed to “black tea; antioxidant activity; catechins; oxidation polymer; theabrownin”.

Point 6  Introduction: Please provide more information regarding the black tea market, including global production and market size ($).

Response 6: More information were supplied according to the suggestion. The sentences “In the year 2020, the global tea production was 6.27 million tons, and it had been growing for recent 10 years (data sources: China tea marketing association).” and “The global tea export and tea market price in major black tea producing countries such as Kenya and Sri Lanka had shown fluctuations.” , were added into the manuscript.

Point 7 Introduction: The sentence “FRAP and DPPH [4] and leaf grade [5] also had difference [6]” is generic. Each determination method is specific for different groups of compounds and are complementary. Review the sentence.

Response 7:  The sentence “FRAP and DPPH [4] and leaf grade [5] also had difference [6]” was changed to “FRAP, total reducing capacity assays and DPPH methods might give different results on the analysis of the same tea samples, and leaf grade and different processing technology also influenced the antioxidant activity of black tea [5-7]”.

Point 8 “47 black tea samples from Yunnan, Sichuan, Fujian, Guizhou, Guangdong, Anhui, Jiangxi, Hubei, Shanxi, Hunan, Henan Province, and 3 black tea samples from Sri Lanka, 1 black tea sample from India.”: Review the sentence, information is missing (were used? Analyzed?).

Response 8:  The sentence was changed to “A total of 51 black tea were analysed in this study. Among that, 47 of them were bought from Yunnan, Sichuan, Fujian, Guizhou, Guangdong, Anhui, Jiangxi, Hubei, Shanxi, Hunan, Henan Province, and 3 black tea samples from Sri Lanka, and 1 black tea sample from India.”

Point 9 2.4. Determination of total antioxidant activity (FRAP method): The use of a single methodology for a study that wants to elucidate antioxidant properties is critical, since each method can detect different chemical groups and components.

Response 9  This study focus on the correlation between catechin and its oxides with antioxidant activity. In our preliminary stuey, the extraction method was optimized, and four different antioxidant activity analysis methods were compared. Finally, this method, FRAP, was selected out for this study because its data was the most stationary.

Point 10 2.9. Determination of total sugar: Change “spectrophotometer, Glucose” to “spectrophotometer. Glucose”

Response 10:  The content has been corrected.

Point 11  Determination of thearubigins (TRs) and Theabrownins (TBs): Change “of 25 mL. the 15 mL ethyl acetate layer” to “of 25 mL. 15 mL ethyl acetate layer”.

Response 11:  The content has been corrected.

Point 12  3.1. Variable coefficient: Change “of a set of data, As shown in Table 1” to “of a set of data. As shown in Table 1”.

Response 12: The content has been corrected.

Point 13  “the oxidation reaction of catechins during black tea fermentation: Deepen the discussion on oxidation.

Response 13: The discussion was modified according to the suggestion. The sentences in 3.4 “Black tea fermentation process is accompanied by a series of oxidation reactions of catechin. Catechin first tranformed into theaflavin and theasinensin through benzophenone ketoneization and disproportionation. These catechin dimers are unstable, and are prone to forming Thearubigin and Theabrownin with oxidation reactions[33]” was added.

Point 14 In addition to variation by fermentation, what other discussions can be observed in relation to the wide variety of composition of chemical components? The discussion must be deepened.

Response 14: The sentences “Although TRs and TBs were also important antioxidant components, and the content were higher than catechins in black tea [12-13], they were not important factors to determine the antioxidant activity of black tea, especially the content of tea TBs was significantly negatively correlated with the antioxidant activity. The mechanism needed further analysis.” was added.

Point 15   Figure 1: The figure resolution is low, and the legend is not visible.

Response 15: The figure has been corrected.

Point 16   3.3. Correlation analysis of: Change “analysis, figure 2.a” to “analysis, Figure 2.a”.

Response 16: The content has been corrected.

Point 17   Figure 2: The figure resolution is low, and the legend is not visible.

Response 17: Thanks for the suggestion. The figure was created online by the website, and the figure resolution can not be adjusted. If necessary, we will supply the original figure by e-mail again later. However, the legend was shown on the right side of the figure, the differences in correlation coefficients represented by different colors.

Point 18  The biochemical aspects of production and degradation of antioxidant compounds must be deepened, given the large variation between samples.

Response 18: Thanks for the suggestion. The new manuscript has interprated the mechanism of catechin oxidation.

Point 19  Correlation analysis of antioxidant activity and chemical compounds during black tea fermentation: Similarly, discussion of changes during fermentation should be highlighted.

Response 19:  Thanks for the suggestion. Correlation analysis of antioxidant activity and chemical compounds during black tea fermentation was mainly used to verify the relationship of catechin and its oxide with antioxidant activity of black tea. However, changes during fermentation was impliedly discussed in the manuscript.

Point 20  Standardize the use of units throughout the text (ml, mL, for example).

Response 20: The content has been corrected.

Reviewer 2 Report

Review notes.

Article title: Analysis of key components on the antioxidant activity of black tea

This manuscript reports the analysis of key components of the antioxidant activity of black tea. The manuscript is well-organized and written. It seems to be a significant contribution to the field of bioactive compounds with antioxidant activity, and it could be published after minor revisions.

The proposed changes are listed below.

Abstract section:

Change “affect” to “influence” in the sentence “It is not clear which components greatly affect the antioxidant activity of black tea”.

Please clarify the meaning of abbreviated words in the abstract section.

Introduction section:

Change “Chinese six major tea” to “six major Chinese”

Change “widely” to “wide” in the first paragraph (Line 7 of the introduction section).

Remove “activity” after antioxidant… (line 1 of the second paragraph)

Change “activity” to “activities” (line 2 of the second paragraph).

Please add a brief description of the benefits to the human health of the antioxidant activity of compounds ¿Why is antioxidant activity one of the most important biological activities of herbal tea?

Remove word “is” in the following sentence “This study is aimed to investigate”…

Materials and methods section:

Please revise that all abbreviations are described the first time that they appear into the document.

Please add a flow chart of the experimental section

Section 2.1

Shengzhou(120◦49′E; 29◦35′N), please add a space between “Shengzhou” and “(120◦49′E; 29◦35′N)”

In the third paragraph, could you add some reference?

Section 2.3

“The tea powder(0.125g)”, please add a space between “The tea powder” and “ (0.125g)”

Section 2.4

reagent 2 , please remove the blank space between “2” and “,”

Please add “and” after reagent 3, reagent 4 as required

Section 2.5

Change comma (UV 3600, Shimadzu), to point (UV 3600, Shimadzu).

Could you provide a reference???

Please add “Results were expressed as mg/g dry sample or wet sample”

Section 2.6

“column(5 μm,..” please add a space between “column” and “(5..”

Could you provide a reference???

Please add “Results were expressed as mg/g dry sample or wet sample”

Section 2.7

Could you provide a reference???

Please add “Results were expressed as mg/g dry sample or wet sample”

Section 2.8

Could you provide a reference???

Please add “Results were expressed as mg/g dry sample or wet sample”

Section 2.9

Change the comma after spectrophotometer to point.

Please add “Results were expressed as mg/g dry sample or wet sample”

Could you provide a reference???

Section 2.10

Could you provide a reference???

Please add “Results were expressed as mg/g dry sample or wet sample”

Results and discussion section:

Section 3.1

Please add how all readers should interpret the variable coefficient to facilitate their meaning in the sentence. For example, the variable coefficients have an interval??

A low coefficient is good??

A high coefficient is good??

Please add a reference at the end of the first paragraph before Table 1. At this point “it was the most variable components among black teas due to different varieties and processing technologies.”

Please add the names of abbreviations included in Table 1 as table foot.

Section 3.2

Please add Figures 1a-e, as separate figures. They are overlapping in the document.

The explained variance is 44.89% in the first two components, is this accurate?? Or is the low or high-variance explanation?

Principal component analysis has been used in other similar studies???, please add a comparative from other studies.

Why use a principal component analysis? Please add a brief description of this statistical tool as a theoretical framework in this section.

Authors have divided the AOX activity of black tea in two classes. The tea samples investigated were clearly divided into two groups: the high antioxidant activity and same moderately antioxidant activity tea samples in class I, the low antioxidant activity and the rest moderately antioxidant activity tea samples in class II. Please add the quantitative AOX activity range for each class. In the same paragraph, please discuss your results and compare them with other authors.

Page 7 at the end. Change “samplers” to “sample”

Section 3.3

This research showed that four phenotypic catechins were positively correlated with the antioxidant activity of black tea, the correlation coefficient ECG and EGCG with black tea antioxidant activity were also higher than EC and EGC, the phenomenon was similar to the previous studies. Please add references.

The authors mentioned that CAF and AA were significant positive correlation with antioxidant activity (section 3.3), but they mentioned that “Although AA and CAF had no antioxidant activity” in section 3.2. Please clarify the meaning of these sentences.

The contents of TF, TF3G, TF3’G and thearubigins were not significantly correlated with antioxidant activity, on the contrary theabrownin was significantly negatively correlated with antioxidant activity. In addition, flavonoids and total sugar were not significantly correlated with antioxidant activity. Please explain, compare with other authors, and discuss about these behaviors.

The correlation coefficient gradually decreased, when thearubigin and theabrownin were increasing, and there was a significant decrease coupling with theabrownin production. What is happening to observe this phenomenon?, please explain, compare with other authors, and discuss.

As mentioned above, catechin and its oxidized products were important substances that determined the antioxidant activity of black tea. Please add a schematic representation of catechin and its biotransformation oxidation/reduction to obtain other polymeric compounds.

Section 3.5

Why ECG exhibited higher values (Figure 5) than the other compounds?. Please explain, compare with other authors, and discuss.

Conclusion section

Please add the practical applications of these results.

Please add some perspectives of this study.

Other notes:

The document did not contain references and number lines. In this context, revising the relevance of the cited references is impossible.

Please minimize the use of abbreviations; excessive use complicates the document‘s readability.

Author Response

The manuscript has been modified according to modification suggestions.

Reviewer 3 Report

Dear Editor,

Dear authors,

The manuscript is well-designed. Numerous experiments were performed, the data were statistically processed and the results were interpreted in a satisfactory manner.

However, it is necessary to improve the following:

·        Correct numerous language and spelling errors throughout the manuscript.

A few examples:

-However, It is not clear which components greatly affect the antioxidant activity of black tea.

-47 black tea samples from Yunnan, Sichuan, Fujian, Guizhou, Guangdong, Anhui,

Jiangxi, Hubei, Shanxi, Hunan, Henan Province, and 3 black tea samples from Sri Lanka,

1 black tea sample from India.

- Analytical reagents including sodium carbonate, Glutamate, Quercetin

rhamnoside, ninhydrin, stannous chloride, oxalic acid, disodium hydrogen phosphate,

potassium dihydrogen phosphate.

- The tea powder(0.125g) was weighed (accurate to 0.0001 g) into 10 mL centrifuge tubes, and extracted for 10 min at 70 ◦C with 5mL 70% aqueous methanol, after centrifugation at 3500 rpm for 10 min, the supernatant were collected into volumetric flask of 10 mL, and the solid was re-extracted by the above process, then two supernatants were combined, and was diluted with 70% methanol into 10 mL, then diluted by 50 times with 70% methanol as the sample extract.

- 3.1. Variablecoefficient. analysis of antioxidant activity and chemical compounds of black tea

·        State that all spectrophotometric measurements were made on the same instrument and not repeat for each experiment. The same applies to the HPLC instrument.

·        The meaning of the abbreviations must be written below each table and/or figure.

·        The resolution of the images is not satisfactory.

·        Conclusion sentence Therefore, it was speculated that the fresh tea leaves had a significant impact on the antioxidant activity of black tea. is unnecessary. It is understood that the quality of black tea depends on the leaves that undergo the production process.

·        The first two highlight items present the same facts.

·         In the fourth item, part was observed in catechin oxides of black tea

 should be omitted.

Dear Editor,

Dear authors,

The manuscript is well-designed. Numerous experiments were performed, the data were statistically processed and the results were interpreted in a satisfactory manner.

However, in order for the manuscript to be published, it is necessary to improve the following:

·        Correct numerous language and spelling errors throughout the manuscript.

A few examples:

-However, It is not clear which components greatly affect the antioxidant activity of black tea.

-47 black tea samples from Yunnan, Sichuan, Fujian, Guizhou, Guangdong, Anhui,

Jiangxi, Hubei, Shanxi, Hunan, Henan Province, and 3 black tea samples from Sri Lanka,

1 black tea sample from India.

- Analytical reagents including sodium carbonate, Glutamate, Quercetin

rhamnoside, ninhydrin, stannous chloride, oxalic acid, disodium hydrogen phosphate,

potassium dihydrogen phosphate.

- The tea powder(0.125g) was weighed (accurate to 0.0001 g) into 10 mL centrifuge tubes, and extracted for 10 min at 70 ◦C with 5mL 70% aqueous methanol, after centrifugation at 3500 rpm for 10 min, the supernatant were collected into volumetric flask of 10 mL, and the solid was re-extracted by the above process, then two supernatants were combined, and was diluted with 70% methanol into 10 mL, then diluted by 50 times with 70% methanol as the sample extract.

- 3.1. Variablecoefficient. analysis of antioxidant activity and chemical compounds of black tea

·        State that all spectrophotometric measurements were made on the same instrument and not repeat for each experiment. The same applies to the HPLC instrument.

·        The meaning of the abbreviations must be written below each table and/or figure.

·        The resolution of the images is not satisfactory.

·        Conclusion sentence Therefore, it was speculated that the fresh tea leaves had a significant impact on the antioxidant activity of black tea. is unnecessary. It is understood that the quality of black tea depends on the leaves that undergo the production process.

·        The first two highlight items present the same facts.

·         In the fourth item, part was observed in catechin oxides of black tea

 should be omitted.

Author Response

(The authors gave the same response as above.)

Round 2

Reviewer 1 Report

Authors have improved the quality of the work.